# More Agents Is All You Need

**Junyou Li**[*]                                    *junyouli@tencent.com*
*Tencent*

**Qin Zhang**[*]                                    *adrienzhang@tencent.com*
*Tencent*

**Yangbin Yu**                                    *yangbinyu@tencent.com*
*Tencent*

**Qiang Fu**                                    *leonfu@tencent.com*
*Tencent*

**Deheng Ye**[†]                                    *dericye@tencent.com*
*Tencent*

**Reviewed on OpenReview:** *https://openreview.net/forum?id=bgzUSZ8aeg*

## Abstract

We find that, simply via a sampling-and-voting method, the performance of large language models (LLMs) scales with the number of agents instantiated. Also, this method, termed as Agent Forest, is orthogonal to existing complicated methods to further enhance LLMs, while the degree of enhancement is correlated to the task difficulty. We conduct comprehensive experiments on a wide range of LLM benchmarks to verify the presence of our finding, and to study the properties that can facilitate its occurrence. Our code is publicly available at: https://github.com/MoreAgentsIsAllYouNeed/AgentForest.

## 1 Introduction

Although large language models (LLMs) demonstrate remarkable capabilities in variety of applications (Zhao et al., 2023), such as language generation, understanding, and reasoning, they struggle to provide accurate answers when faced with complicated tasks. To improve the performance of LLMs, some of recent studies focus on ensemble methods (Wang et al., 2023b; Wan et al., 2024) and multiple LLM-Agents collaboration frameworks (Du et al., 2023; Wu et al., 2023).

In these works, multiple LLM agents are used to improve the performance of LLMs. For instance, LLM-Debate Du et al. (2023) employs multiple LLM agents in a debate form. The reasoning performance is improved by creating a framework that allows more than one agent to "debate" the final answer of arithmetic tasks. They show performance improvements compared to using one single agent. Similarly, CoT-SC (Wang et al., 2023b) generates multiple thought chains and picks the most self-consistent one as the final answer. The reasoning performance is improved by involving more thought chains compared to chain-of-thought (CoT) (Wei et al., 2022) which employs a single thought chain. Incidentally, from the data analysis of these works, we can notice the effects of putting multiple agents together, to some extent, can lead to a performance improvement in certain problems. For example, in Table 10 of Section 3.3 of LLM-Debate Du et al. (2023), the authors have reported a preliminary curve: the accuracy of a math problem increases with the number of debating agents (although the number was simply increased from 1 to 7). Also, in Wang et al. (2023b), involving more chain-of-thought pipelines (termed as a "sample-and-marginalize" decoding procedure), can

---

[*]Co-first authors.
[†]Corresponding author.

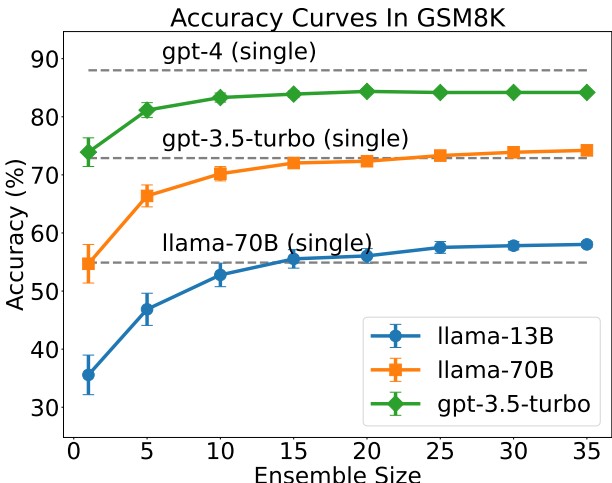

Figure 1: The accuracy increases with ensemble size across Llama2-13B, Llama2-70B and GPT-3.5-Turbo in GSM8K. When the ensemble size scales up to 15, Llama2-13B achieves comparable accuracy with Llama2-70B. Similarly, When the ensemble size scales up to 15 and 20, Llama2-70B and GPT-3.5-Turbo achieve comparable accuracy with their more powerful counterparts. The error bars represent the standard error.

lead to a performance gain. We realize that the LLM performance may likely be improved by a brute-force scaling up of the number of agents instantiated. However, since the scaling property of "raw" agents is not the focus of these works, the scenarios/tasks and experiments considered are limited. So far, there lacks a dedicated in-depth study on such a phenomenon. Hence, a natural question arises: *Does this phenomenon generally exist?*

To answer the research question above, we conduct the first comprehensive study on the *scaling property* of LLM agents. To dig out the potential of multiple agents, we propose to use a simple(st) sampling-and-voting method, which involves two phases. First, the query of the task, i.e., the input to an LLM, is iteratively fed into a single LLM, or a multiple LLM-Agents collaboration framework, to generate multiple outputs. Subsequently, majority voting is used to determine the final result. The procedure is inspired by that of the CoT-SC, but it does not rely on designing complex CoT paths. In fact, it can be used as a plug-in to further enhance CoT-based methods, as will be shown in our evaluations. Our method is termed as **Agent Forest**, a tribute to the classic Random Forest (Breiman, 2001).

The experiments are conducted by using various LLMs of different sizes on diverse datasets covering reasoning and generation. The result indicates that LLM performance can generally be improved by increasing the ensemble size, i.e., the number of agents, across a wide range of tasks. Surprisingly, a brute-force ensemble of smaller LLMs can achieve comparable or superior performance to larger LLMs, with a nutshell shown in Figure 1, which will be further expanded in later sections. Moreover, by combining our method with other existing methods, we find the performance can be further improved. By comparing with the performance of complicated methods, the result shows that employing our method solely can achieve comparable performance in most cases. This implies that comparable performance can be achieved without the need for additional handcraft prompt design or complex collaboration frameworks.

Additionally, the experimental results indicate that there are greater performance improvements when addressing difficult tasks and when using weaker models. To understand the reasons behind these performance improvements, we analyze the influence of problem difficulty on the effectiveness of our method. We classify difficulty into three dimensions: the inherent difficulty, the length of reasoning steps, and the prior probability of the correct answer. Through a series of experiments, we adjust these dimensions and observe their effects independently. We observe and summarize a few properties, based on which, we further develop optimization strategies that can intrigue the power of "More Agents".

Our contributions are summarized as follows:

- We present the first systematic study on the scaling property of raw agents instantiated by LLMs. We find that the performance scales with the increase of agents, using the simple(st) way of sampling and voting.

- We explore the compatibility of our method with existing complicated methods that stimulate the potential of LLMs, revealing that our method can enhance these methods to achieve further performance improvements.

- We analyze the effectiveness of our method in tackling problems at varying difficulties and then distill the properties behind, based upon which, we propose further optimization methods that can facilitate the occurrence of our finding.

## 2 Related Work

Related works can be categorized into three parts: 1) LLM self-ensemble Wang et al. (2023b), which attempts to harness multiple outputs from homogeneous LLMs to assemble the final answer; 2) heterogeneous LLM ensemble, which focuses on combining heterogeneous LLMs through supervised learning to improve performance across various downstream applications; and 3) multiple LLM agents collaboration, which improves performance through interactions among LLM agents. We discuss these works below.

**LLM Self-Ensemble.** CoT-SC Wang et al. (2023b) harnesses diverse chain-of-thought Wei et al. (2022) prompts to elicit a variety of reasoning processes from a single LLM and select the final answer through majority voting. Fu et al. (2023); Li et al. (2023b); Cobbe et al. (2021b); Thoppilan et al. (2022); Lin et al. (2023) can be considered as the extensions of CoT-SC. These methods mainly focus on reasoning tasks and exclusively investigate the compatibility with CoT. In contrast, our method not only validates effectiveness in reasoning tasks but also in generation tasks. Moreover, our method is compatible with a broader range of methods, such as prompt engineering (including CoT) and multiple LLM agents collaboration. Very recently, Lu et al. (2024) proposes a method named Blended that utilizes multiple LLMs for chat scenarios. In contrast, Blended focuses on utilizing the power of multiple LLMs, whereas our focus is on the scaling trend of adding more LLMs. Also, Blended is only for limited chat scenarios evaluated via human annotations. Furthermore, we explore orthogonality with other methods.

**Heterogeneous LLM Ensemble.** Wan et al. (2024) conducts a supervised LLM fusion framework to distill multiple heterogeneous LLMs into a single model and surpasses each of these LLMs. Jiang et al. (2023) introduces a supervised ensembling framework based on multiple heterogeneous LLMs. Chen et al. (2023b) proposes a sequential inference method for LLMs that halts when the output quality is deemed adequate. Wang et al. (2023a) addresses the fusion-of-experts problem by integrating outputs from models with distinct knowledge domains through supervised learning. Shnitzer et al. (2023) and Lu et al. (2023) select the most suitable LLM for new tasks by training a reward-guided router. These approaches primarily employ supervised learning, necessitating task-specific annotated data, and exhibit limited generalizability. In contrast, our method is unsupervised, without the need for additional training data.

**Multiple LLM Agents Collaboration.** Du et al. (2023); Liang et al. (2023); Xiong et al. (2023) explore various multiple LLM agents interaction architectures, with employing static debate-style engagements among LLMs for enhanced reasoning . Liu et al. (2023) enables agents to interact for multiple rounds in a dynamic architecture. Li et al. (2023a); Hong et al. (2023); Wu et al. (2023); Chen et al. (2023c;a) offer several multi-agent frameworks that enable the development of LLM applications or enhance task-solving capabilities. However, these methods primarily focus on the interaction structures between LLM agents, rather than the relationship between the number of agents and performance. We also select representative methods Du et al. (2023); Shinn et al. (2023) to combine with our method, achieving further enhancements.

## 3 Method

In this section, we introduce **Agent Forest**, which is implemented through a two-phase process: sampling and voting. The overview of our method is shown in Figure 2.

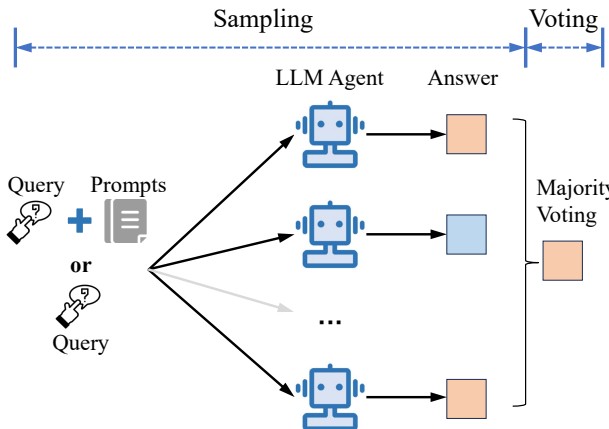

Figure 2: Illustration of Agent Forest. The two-phase process begins by feeding the task query, either alone or combined with prompt engineering methods, into LLM agents to generate answers. Subsequently, majority voting is applied to these answers to determine the final answer. Specifically, an LLM agent refers to a single LLM or a multiple LLM-Agents collaboration framework.

---

**Algorithm 1** Agent Forest

---

**Require:** Query $x$, number of samples $N$, LLM $\mathcal{M}$ or LLM integrated with other methods $f_{\mathcal{M}}(x)$
1: Initialize an empty set for samples $S \leftarrow \emptyset$
2: **for** $i = 1$ to $N$ **do**
3:     Generate sample $s_i \leftarrow \mathcal{M}(x)$ or $s_i \leftarrow f_{\mathcal{M}}(x)$
4:     Add sample to the set $S \leftarrow S \cup \{s_i\}$
5: **end for**
6: **for** each sample $s_i$ in $S$ **do**
7:     Initialize similarity scores $V(s_i) \leftarrow 0$
8:     **for** each sample $s_j$ in $S$ **do**
9:         **if** $i \neq j$ **then**
10:           $V(s_i) \leftarrow V(s_i) + sim(s_i, s_j)$
11:         **end if**
12:     **end for**
13: **end for**
14: $A \leftarrow \arg\max_{s_i \in S} V(s_i)$
15: **return** $A$

---

**Sampling.** Let $x$ represent the task query and $\mathcal{M}$ denote an LLM. In this phase, we generate $N$ samples by solely querying the LLM $\mathcal{M}$ $N$ times with each sample represented as $s = \mathcal{M}(x)$ or by integrating with other methods $f_{\mathcal{M}}$ with $N$ times executions where each sample is denoted as $s = f_{\mathcal{M}}(x)$. We obtain a set of samples $S = \{s_1, s_2, ..., s_N\}$ at the end of this phase.

**Voting**. Let $A$ represent the final answer. In this phase, we employ majority voting to consolidate the response sample set $S$ into the final answer $A$. This involves calculating the cumulative similarity for each sample relative to the others, denoted as $V(s_i) = \sum_{j=1, j\neq i}^{N} sim(s_i, s_j)$. For open-ended generation tasks such as code generation, the BLEU score proposed by Papineni et al. (2002) is utilized to quantify similarity. Conversely, for close-ended tasks like multiple-choice questions, similarity is measured by occurrence frequency. The sample that exhibits the highest cumulative similarity is then chosen as the final answer denoted as $A = \arg\max_{s_i \in S} V(s_i)$.

The complete process of Agent Forest is described in Algorithm 1.

## 4    Experimental Setup

We separate the experimental setup (this section) with evaluations (next section), to introduce the coverage of scenarios/tasks compared with the most related works (for examining the comprehensiveness of our work), the backbone language models we adopted (for examining the applicability of our work), and the methods combined with ours (for examining the compatibility and orthogonality of our work).

**Tasks**    Our method is evaluated on the following task:

- Arithmetic Reasoning. Similar to Wang et al. (2023b); Fu et al. (2023); Du et al. (2023), we select the GSM8K Cobbe et al. (2021a) as one of the test sets. Additionally, we select the more challenging MATH dataset Hendrycks et al. (2021b), which is used by Wu et al. (2023).

- General Reasoning. Similar to Du et al. (2023); Jiang et al. (2023), we select the MMLU Hendrycks et al. (2021a). Additionally, we select the dataset from the chess state tracking task (Chess) [1], which is used by Du et al. (2023); Zhang et al. (2023).

- Code Generation. Similar to Liu et al. (2023), we select the HumanEval Chen et al. (2021). To implement our method, we compute the BLEU score Papineni et al. (2002) among all pairs of generated candidate answers. The answer with the highest cumulative BLEU score is then selected as the final output.

Table 1: Comparing the conducted experiments with the most related works. Our comprehensive study encompasses various LLMs, multiple tasks, and the integration with multiple methods.

| Methods | Various LLMs | Tasks | | | | Integrated with Methods | |
| --- | --- | --- | --- | --- | --- | --- | --- |
| | | Chat | Arithmetic Reasoning | General Reasoning | Code Generation | Prompt Engineering | Multiple LLM-Agents Collaboration |
| CoT-SC Wang et al. (2023b) | ✓ | | ✓ | ✓ | | Only CoT Wei et al. (2022) | |
| Complexity-CoT Fu et al. (2023) | ✓ | | | ✓ | | Only CoT Wei et al. (2022) | |
| Debate Du et al. (2023) | | | ✓ | | | | |
| Blended Lu et al. (2024) | ✓ | ✓ | | | | | |
| Ours | ✓ | | ✓ | ✓ | ✓ | ✓ | ✓ |

**Language models adopted**    We evaluate our method using language models of different scales from the Llama2 Touvron et al. (2023) and GPT series OpenAI (2022). Specifically, we evaluate two versions of Llama2-Chat[2], optimized for conversational use cases through alignment techniques, with model sizes of 13B and 70B parameters. Additionally, we include GPT-3.5-Turbo and GPT-4 in our evaluation.

**Methods enhanced by our method**    To examine the comparability of our method, we study the integration of various typical methods from two distinct categories with our method:

- Prompt Engineering. Various prompt engineering methods are considered to conduct comprehensive experiments. We evaluate Chain-of-Thought prompting (CoT) Wei et al. (2022), Zero-Shot Chain-of-Thought prompting (Zero-Shot Cot) Kojima et al. (2022), and more sophisticated methods such as Solo Performance Prompting (SPP) Wang et al. (2023c). Initially, these methods are applied with a single LLM query. We then increase the number of queries and employ majority voting to determine the most consistent answer as the final response.

- Multiple LLM Agents Collaboration. We select LLM-Debate Du et al. (2023) denoted as Debate, and self-reflection Shinn et al. (2023) denoted as Reflection. Within these methods, we generate multiple samples by iteratively operating these methods and using majority voting to produce the final answer.

---

[1]Chess State Tracking
[2]Llama2-Chat

Specifically, the effectiveness of our method is evaluated by averaging the results across 10 independent runs. During each run, we scale up the ensemble size to 40 to ensure maximum gains. However, when integrating our method with the Debate Du et al. (2023), the ensemble size is limited to 10 due to the significant computational overhead introduced by the communication architecture. Detailed experimental settings are provided in the Appendix A.

## 5 Experimental Results

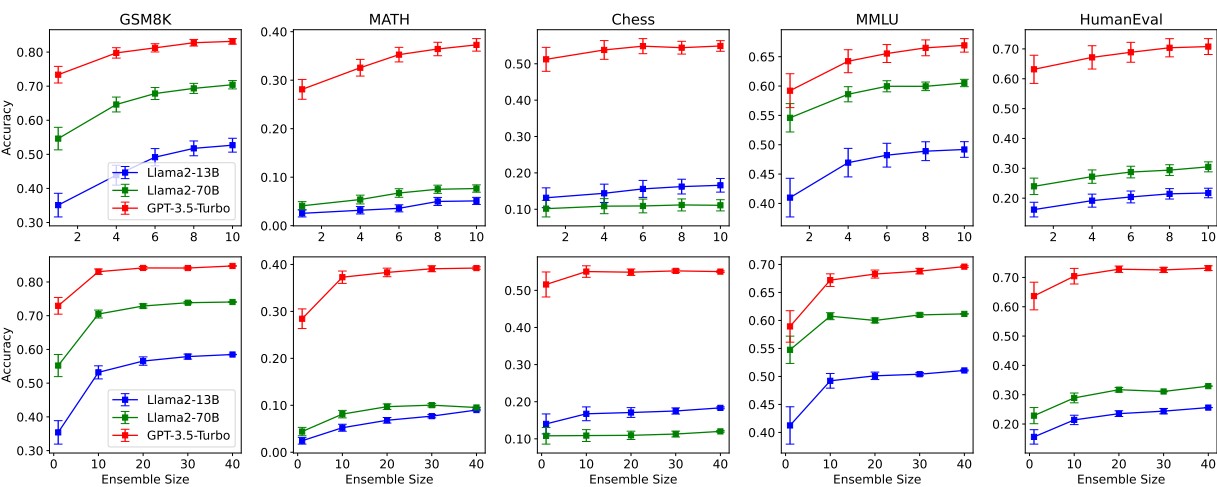

Figure 3: The accuracy scales with the ensemble size of our method across different tasks with various LLMs. The error bars represent the standard error.

Table 2: Our method generally enhances performance across all tasks and LLMs. The bolded instances indicate that smaller LLMs outperform the larger LLMs. "Single" denotes that the LLM is queried only once. GPT-4 is used only for comparison with other methods, hence it only presents "Single" results. "Ours" denotes our method where the ensemble size is 40. The error bars represent the standard error.

| Model | GSM8K | | MATH | | Chess | | MMLU | | HumanEval | |
|---|---|---|---|---|---|---|---|---|---|---|
| | Single | Ours | Single | Ours | Single | Ours | Single | Ours | Single | Ours |
| Llama2-13B Touvron et al. (2023) | 0.35 ± 3e-2 | **0.59** ± 5e-4 | 0.03 ± 7e-3 | **0.09** ± 2e-3 | 0.14 ± 2e-2 | **0.18** ± 2e-3 | 0.42 ± 3e-2 | 0.51 ± 1e-3 | 0.14 ± 1e-2 | 0.18 ± 1e-3 |
| Llama2-70B Touvron et al. (2023) | 0.54 ± 3e-2 | **0.74** ± 1e-3 | 0.05 ± 1e-2 | 0.11 ± 1e-3 | 0.12 ± 2e-2 | 0.13 ± 2e-3 | 0.55 ± 2e-2 | **0.60** ± 3e-3 | 0.24 ± 1e-2 | 0.33 ± 1e-3 |
| GPT-3.5-Turbo OpenAI (2022) | 0.73 ± 2e-2 | 0.85 ± 3e-3 | 0.29 ± 2e-2 | 0.39 ± 2e-3 | 0.51 ± 3e-2 | 0.55 ± 2e-3 | 0.59 ± 3e-2 | 0.70 ± 2e-3 | 0.67 ± 2e-2 | 0.73 ± 1e-2 |
| GPT-4 OpenAI (2022) | 0.88 ± 2e-2 | — | 0.40 ± 3e-2 | — | 0.65 ± 2e-2 | — | 0.77 ± 2e-2 | — | 0.88 ± 3e-2 | — |

### 5.1 Generalizability

Table 2 and Figure 3 show that our method generally enhances performance across all tasks and LLMs by increasing the ensemble size. Specifically, in arithmetic reasoning tasks, the accuracy gains range from 12% to 24% on the GSM8K and from 6% to 10% on the MATH. In general reasoning tasks, the accuracy gains range from 1% to 4% on the Chess and from 5% to 11% on the MMLU. In code generation task, the accuracy gains range from 4% to 9% on HumanEval. Surprisingly, our method enables a smaller LLM to outperform a larger counterpart by simply scaling up the ensemble size. For instance, the enhanced Llama2-13B model achieves 59% accuracy on the GSM8K dataset, outperforming the Llama2-70B model, which scores 54%. Additional statistical results are presented in Appendix B.2.

Table 3: Our method outperforms other methods used standalone in most cases and always enhances other methods across various tasks and LLMs. The bolded instances indicate the highest accuracy for each task and the underlined instances indicate the highest accuracy in standalone cases.

| Model | Method | GSM8K | | MATH | | Chess | | MMLU | | HumanEval | |
|---|---|---|---|---|---|---|---|---|---|---|---|
| | | Standalone | +Ours | Standalone | +Ours | Standalone | +Ours | Standalone | +Ours | Standalone | +Ours |
| Llama2-13B Touvron et al. (2023) | COT Wei et al. (2022) | 0.39 | 0.56 (+0.17) | 0.04 | 0.06 (+0.02) | 0.18 | 0.23 (+0.07) | 0.42 | 0.43 (+0.01) | 0.13 | 0.20 (+0.07) |
| | ZS-COT Kojima et al. (2022) | 0.40 | **0.61** (+0.21) | 0.03 | 0.08 (+0.05) | 0.15 | 0.20 (+0.05) | 0.42 | 0.48 (+0.06) | 0.15 | 0.22 (+0.07) |
| | SPP Wang et al. (2023c) | 0.19 | 0.42 (+0.23) | 0.01 | 0.04 (+0.03) | 0.21 | **0.26** (+0.05) | 0.32 | **0.53** (+0.21) | 0.03 | 0.08 (+0.05) |
| | Debate Du et al. (2023) | 0.38 | 0.48 (+0.10) | 0.05 | 0.07 (+0.02) | 0.18 | 0.19 (+0.01) | 0.37 | 0.39 (+0.02) | 0 | 0 |
| | Reflection Shinn et al. (2023) | 0.36 | 0.59 (+0.23) | 0.01 | 0.03 (+0.02) | 0.13 | 0.19 (+0.06) | 0.45 | 0.50 (+0.05) | 0.06 | 0.13 (+0.07) |
| | Ours | 0.59 | | **0.09** | | 0.18 | | 0.51 | | **0.25** | |
| Llama2-70B Touvron et al. (2023) | COT Wei et al. (2022) | 0.57 | 0.72 (+0.15) | 0.06 | **0.13** (+0.07) | 0.10 | 0.11 (+0.01) | 0.56 | 0.57 (+0.01) | 0.30 | 0.32 (+0.02) |
| | ZS-COT Kojima et al. (2022) | 0.57 | 0.73 (+0.16) | 0.04 | 0.10 (+0.06) | 0.20 | **0.27** (+0.07) | 0.54 | **0.65** (+0.11) | 0.23 | 0.29 (+0.06) |
| | SPP Wang et al. (2023c) | 0.42 | 0.69 (+0.27) | 0.03 | 0.09 (+0.06) | 0.16 | **0.27** (+0.11) | 0.49 | 0.63 (+0.14) | 0.15 | 0.20 (+0.05) |
| | Debate Du et al. (2023) | 0.59 | 0.65 (+0.06) | 0.10 | 0.11 (+0.01) | 0.14 | 0.17 (+0.03) | 0.56 | 0.58 (+0.02) | 0 | 0 |
| | Reflection Shinn et al. (2023) | 0.52 | **0.77** (+0.25) | 0.02 | 0.05 (+0.03) | 0.15 | 0.26 (+0.11) | 0.42 | 0.55 (+0.13) | 0.16 | 0.26 (+0.10) |
| | Ours | 0.74 | | 0.11 | | 0.13 | | 0.60 | | **0.33** | |
| GPT-3.5-Turbo OpenAI (2022) | COT Wei et al. (2022) | 0.74 | 0.84 (+0.10) | 0.28 | **0.41** (+0.13) | 0.50 | 0.55 (+0.05) | 0.61 | 0.64 (+0.03) | 0.70 | **0.75** (+0.05) |
| | ZS-COT Kojima et al. (2022) | 0.74 | **0.88** (+0.14) | 0.25 | 0.40 (+0.15) | 0.35 | 0.48 (+0.13) | 0.58 | 0.69 (+0.11) | 0.67 | 0.74 (+0.07) |
| | SPP Wang et al. (2023c) | 0.70 | 0.83 (+0.13) | 0.26 | 0.39 (+0.13) | 0.37 | 0.54 (+0.17) | 0.53 | 0.68 (+0.15) | 0.57 | 0.64 (+0.07) |
| | Debate Du et al. (2023) | 0.83 | 0.85 (+0.02) | 0.32 | 0.36 (+0.04) | 0.49 | **0.57** (+0.08) | 0.56 | 0.67 (+0.11) | 0.18 | 0.24 (+0.06) |
| | Reflection Shinn et al. (2023) | 0.76 | 0.84 (+0.08) | 0.27 | **0.41** (+0.14) | 0.44 | **0.57** (+0.13) | 0.39 | 0.44 (+0.05) | 0.58 | 0.73 (+0.15) |
| | Ours | 0.85 | | 0.39 | | 0.55 | | **0.70** | | 0.73 | |

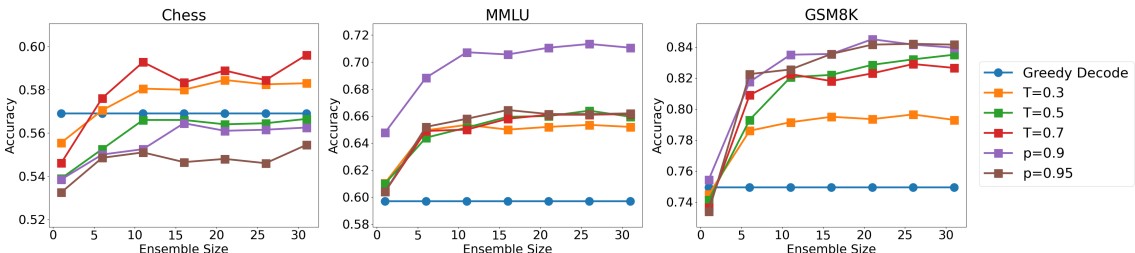

Figure 4: Our method improves accuracy over various hyperparameters and tasks. The default $T$ is 1.0 and the default $p$ is 1.0.

## 5.2 Compatibility

Table 3 shows that by integrating our method with other methods, the performance can be further improved across different LLMs and tasks, despite these methods have different implementations. To be specific, in arithmetic reasoning tasks, our method enhances these methods to further improvement, yielding increases between 10% and 21% on the GSM8K dataset, and between 1% and 15% on the MATH dataset. In general reasoning tasks, integration with other methods generally achieves performance gains ranging from 1% to 13% in the Chess task and from 1% to 11% in the MMLU task. In code generation task, when combined with other methods, gains range from 2% to 7%. However, two notable exceptions are observed when integrated with the debate method with the Llama2-13B and Llama2-70B models, which result in failed cases. This failure in performance is attributed primarily to the noise generated by referencing the answers of other agents during the debate process. The synthesized responses, which incorporate input from multiple agents, disrupt the coherence of the code logic, leading to the observed performance degradation. All accuracy curves are provided in the Appendix B.1.

## 5.3 Effectiveness

From Table 3, we find that our method outperforms other methods in standalone cases, except on the Chess task using Llama2-13B and Llama2-70B. Additionally, based on the data from Table 3, we have calculated the average performance ranking of each enhanced method across various tasks, with the results presented in

Table 4. Notably, without the need for additional prompts or complex LLM collaboration frameworks, our method achieves the highest average ranking across different LLMs and tasks.

Table 4: Our method achieved the highest average ranking across different LLMs and tasks. Rankings are derived from Table 3 and are based on the average rank each method achieves across all five tasks for a given LLM. The bolded instances indicate the top ranking.

| Method +Ours | GPT-3.5 | 70B | 13B | Overall |
|---|---|---|---|---|
| COT Wei et al. (2022) | 2.8 | 3.6 | 3.6 | 3.3 |
| ZS-COT Kojima et al. (2022) | 2.8 | **2.4** | 3 | 2.7 |
| SPP Wang et al. (2023c) | 4.6 | 3.6 | 3.8 | 4 |
| Debate Du et al. (2023) | 3.8 | 4.4 | 5 | 4.4 |
| Reflection Shinn et al. (2023) | 3 | 4.0 | 3 | 3.3 |
| Ours | **2.6** | 2.6 | **2.2** | **2.5** |

## 5.4 Robustness

We conduct ablation studies to evaluate the impact of changes in various hyperparameters on the final performance. The experiment is conducted by altering the temperature $T$ Ficler & Goldberg (2017) and the nucleus probability $p$ Radford et al. (2019), using the GPT-3.5-Turbo model over an average of 20 runs. As shown in Figure 4, scaling up ensemble size improves the LLM performance consistently across different tasks, despite the variation of these hyperparameters.

## 5.5 Token usage

We record the token usage for different methods, with the Table 5 presenting the token usage for a single agent. Given that our method scales up by increasing the ensemble size, the token usage increases proportionally with the number of agents or when combined with other methods. When addressing specific tasks, one can trade a higher token budget for improved performance. More details are presented in Appendix B.3

Table 5: Token usage of GPT-3.5

| Method | GSM8K | MATH | Chess | MMLU | HumanEval |
|---|---|---|---|---|---|
| Ours (single agent) | 235 ± 54 | 326 ± 131 | 138 ± 14 | 247 ± 91 | 495 ± 120 |
| COT Wei et al. (2022) | 261 ± 63 | 360 ± 134 | 284 ± 76 | 330 ± 110 | 330 ± 116 |
| ZS-COT Kojima et al. (2022) | 228 ± 56 | 305 ± 122 | 132 ± 12 | 230 ± 92 | 305 ± 132 |
| SPP Wang et al. (2023c) | 341 ± 85 | 471 ± 161 | 363 ± 35 | 428 ± 123 | 501 ± 125 |
| Reflection Shinn et al. (2023) | 214 ± 45 | 281 ± 96 | 146 ± 13 | 218 ± 80 | 338 ± 110 |

## 6 Understanding the Performance Gains

Table 2 shows that the efficacy of our method varies with the difficulty of the task. In this section, we aim to understand the underlying properties through controlled experiments.

To start the analysis, we select two datasets with increasing difficulty, i.e., GSM8K and MATH, to calculate the relative performance gain. The relative performance gain $\eta$ is given by: $\eta = \frac{P_m - P_s}{P_s}$ where $P_m$ and $P_s$ are the performances (accuracy) with our method and a single LLM query, respectively. The results are shown in Table 6.

It is noteworthy that the relative performance gain is more substantial with increasing task difficulty. Specifically, we observe that within the same task, the smaller model, Llama2-13B, gains ranging from 28%-200%, but only 8%-16% over GPT-3.5-Turbo. Moreover, the more challenging task MATH yields gains of 34%-200%, in contrast to only 16%-69% on the easier task GSM8K.

Table 6: The relative performance gain (%) becomes more significant when the relative difficulty between the LLM and the task increases. It is calculated based on Table 2.

| Task | Llama2-13B | Llama2-70B | GPT-3.5-Turbo |
|------|-----------|-----------|--------------|
| GSM8K (easy) | 69 | 37 | 16 |
| MATH (hard) | 200 | 120 | 34 |

To further analyze this correlation in detail, we categorize the difficulty of a given task into three orthogonal dimensions: 1) the inherent difficulty of the task; 2) the number of steps required to solve the task; 3) the prior probability of the correct answer. To investigate these dimensions, we conduct experiments that can isolate each dimension. And then, we delve into each dimension in detail.

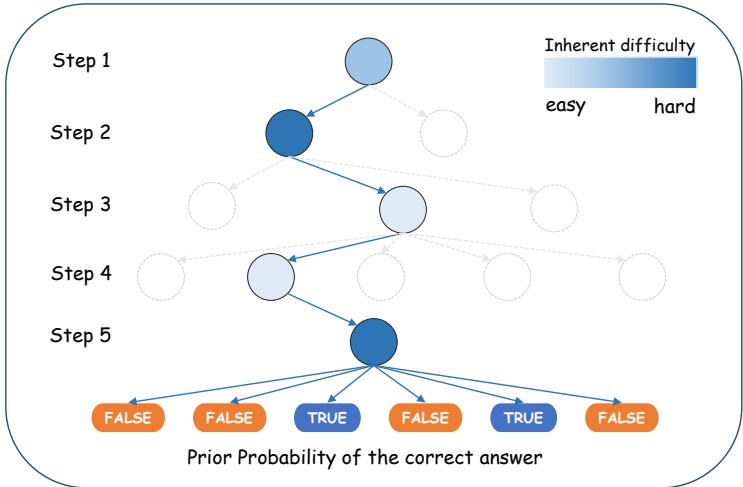

Figure 5: Illustration of three dimensions for a given task. Nodes represent steps, while dashed lines indicate alternative potential steps. The depth of nodes represents the number of steps, and the color intensity represents the level of inherent difficulty.

## 6.1 Isolation

To explicitly explore the impact of these dimensions, we conduct a mathematical task designed to isolate each one. Consider the task detailed below:

$$\text{Find the interval } \Delta_k \text{ such that } \sum_{i=1}^{S} a_i \cdot b_i \in \Delta_k, \tag{1}$$

where:

- $a_i, b_i$ are randomly chosen integers from the closed interval $[-I, I]$. $I \in \mathbb{Z}^+$ defines the range of integers. $I$ represents the **inherent difficulty** of the question. A larger value of $I$ indicates a more challenging task.

- $S \in \mathbb{Z}^+$ is the number of terms in the summation. $S$ represents the **number of steps** required to solve the problem. A larger value of $S$ indicates a more challenging task.

- The result space is partitioned into $K$ intervals $\Delta_1, \Delta_2, \ldots, \Delta_K$ of equal probability. $K \in \mathbb{Z}^+$ denotes the number of these intervals. $1/K$ represents the **prior probability** of the correct answer. A lower prior probability indicates a more challenging task.

In the following experiments, we analyze each dimension respectively based on GPT-3.5-Turbo. Note that we use GPT-3.5-Turbo for a case study, it can also be changed to other backbone models. The relative performance gains are measured by the difference between the maximum accuracy our method can achieve (sampling 40 times) and the accuracy of a single LLM query (sample once). Results are averaged over 10 runs.

## 6.2 Inherent Difficulty

**Property 1**: *Gains increase then decrease by rising the inherent difficulty.* We investigate the inherent difficulty by varying $I$ from 10 to 400, while keeping the values of $S$ and $K$ constant across four groups of different values, from small to large, respectively. Figure 6 (left) shows an initial uptick in performance gains with increases in $I$, indicating that our method can significantly enhance performance in line with rising inherent difficulty. The most notable gains are seen at $I = 100$ and $I = 200$, consistent across all $S$ and $K$ settings. Yet, at $I = 400$, gains taper off, implying that excessive complexity may exceed the model's reasoning capabilities, leading to diminishing returns for our method under extreme task difficulty.

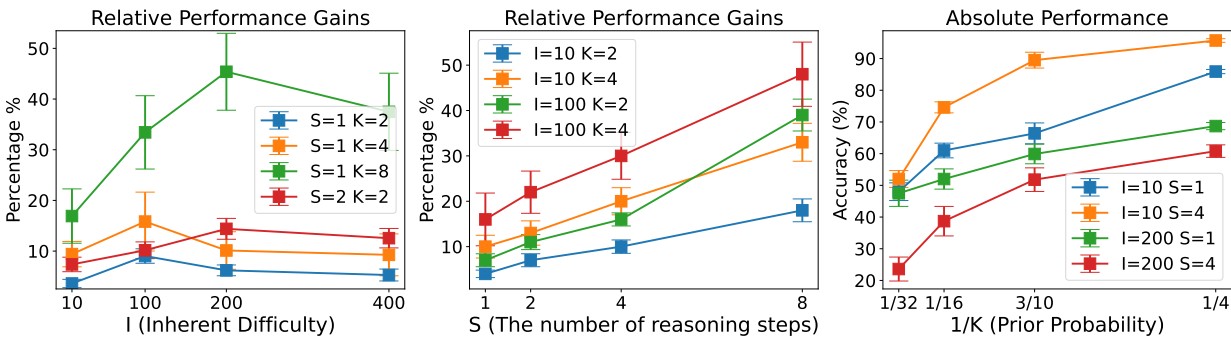

Figure 6: (Left) The relative performance gains increase and then decrease with rising inherent difficulty. (Middle) The relative performance gains increase with the number of steps. (Right) The absolute performance increases with the prior probability. We analyze each dimension by fixing the other two dimensions. The error bars represent the standard error.

## 6.3 Number of Steps

**Property 2.1**: *Gains increase with the number of steps.* We analyze the number of steps by isolating $S$. We tune $S$ from 1 to 8, while keeping the values of $I$ and $K$ constant across four groups of different values, ranging from small to large, respectively. Figure 6 (middle) shows that as the number of steps increases, there is a corresponding increase in performance gain. Additionally, we find that when $I$ and $K$ are increased (which indicates a higher difficulty), the performance gains are more significant, e.g., 4%-18% gains over $\{I = 10, K = 2\}$ compared to 16%-48% over $\{I = 100, K = 4\}$.

**Property 2.2**: *Agent Forest increases the performance for each step.* We conduct a fine-grained analysis for each step of a given task. We explicitly prompt the language model to output the result of each step. Subsequently, we utilize Agent Forest at each step to derive the answer for that step. Figure 7 (left) shows that although each step has equal inherent difficulty, the accumulation of errors from previous steps lead to a decrease in accuracy as the number of steps increases. However, our method mitigates the performance decrease encountered with increasing steps.

**Derivation.** Based on Property 2, we propose Step-wise Agent Forest, which can further enhance the performance.

Step-wise Agent Forest initially prompts the LLM to decompose the task into multiple steps. It then proceeds with multi-round iterations to produce the final result. In each round, the process begins by selecting a current unprocessed step and using Agent Forest to determine the result of that step. Subsequently, it uses

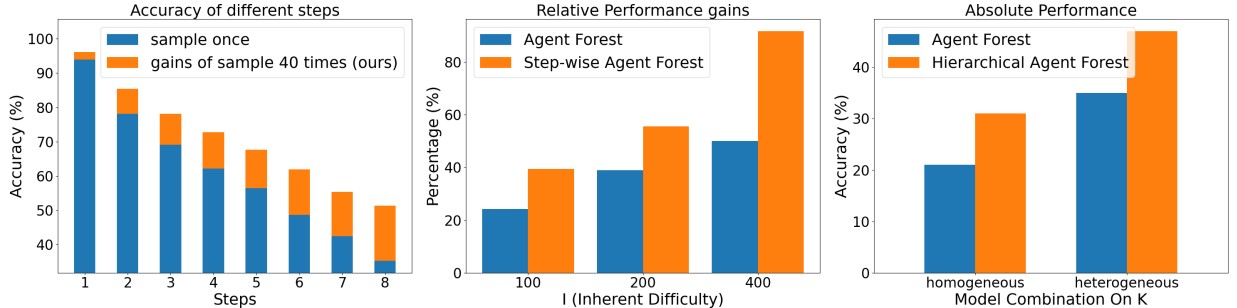

Figure 7: (Left) Our method increases the performance for each step. Blue bars show the accuracy of various steps for a single sample, and orange bars show the gains for 40 samples. (Middle) Step-wise Agent Forest can further enhance the performance across different levels of inherent difficulty. (Right) Hierarchical Agent Forest can further enhance the performance with homogeneous and heterogeneous model combinations.

the result to update the task. This iterative process is repeated multiple times until the last step is processed. To evaluate the performance of Step-wise Agent Forest, we fix $S = 8$ and $K = 4$, and tune $I$ from 100 to 400. Figure 7 (middle) shows that compared to Agent Forest, Step-wise Agent Forest yields greater improvements. e.g., we see 15%-42% gains, which increase with inherent difficulty.

## 6.4 Prior Probability

`Property 3`: *The performance increases with the prior probability.* We investigate the influence of prior probability on performance by tuning the parameter $K$, while maintaining constant values for $I$ and $K$. As $K$ represents the number of intervals, the prior probability is defined as $1/K$. We vary $K$ from 4 to 32, which equivalently alters the prior probability from 1/4 to 1/32. Through four experimental groups illustrated in Figure 6 (right), each characterized by different configurations of $I$ and $S$, we find that as the prior probability increases, so does the performance.

`Derivation.` Based on Property 3, we propose Hierarchical Agent Forest can further enhance the performance.

As the performance is related to the prior probability, decomposing low-probability tasks into multiple high-probability subtasks and addressing them hierarchically can boost performance. Moreover, subtasks with varying prior probabilities can be addressed using different models. Additionally, cost savings can be achieved by using simpler, less expensive models for the easier, higher-probability subtasks.

In our experiments, the task is to solve the problem with $K = 32$. GPT-3.5-Turbo is used in homogeneous combination experiment and GPT-3.5-Turbo and GPT-4 are used in heterogeneous combination experiment. The results are presented in Figure 7 (right).

In homogeneous combination experiment, by employing the hierarchical method, we start with $K = 8$ to obtain an intermediate answer and then find the solution with $K = 32$, focusing on intervals identified by the intermediate answer. This method enhances the performance from 21% to 31%, demonstrating that the hierarchical method can further enhance the performance.

In heterogeneous combination experiment, GPT-3.5-Turbo is used for generating the intermediate answer with $K = 8$, and GPT-4 is then employed to solve for the final answer with $K = 32$. In Figure 7 (right), compared with the result of GPT-4 with $K = 32$, the hierarchical method improves performance from 35% to 47%, suggesting the deployment of different LLMs at the corresponding level of problem-solving can improve the performance in a cost-effective manner.

## 7 Conclusions and Future Work

In this paper, we report that *more agents is all you need*, i.e., simply adding more instantiated LLM agents is what you need to obtain a better LLM performance in processing complex tasks, without bothering complicated methods, such as CoT pipelines, multi-agent collaboration frameworks, etc. We have conducted the first comprehensive study in the literature to understand such a "scaling law", including when it holds and how to facilitate its occurrence.

The results indicate that Agent Forest can generally improve the performance of LLMs by increasing the ensemble size. Importantly, this method is orthogonal to different existing methods, which can lead to further improvements when combined with them.

Furthermore, we observe that the performance gains are influenced by the difficulty of the task. To explore this correlation, we isolate and analyze three dimensions of task difficulty: the inherent difficulty, the length of reasoning steps, and the prior probability of a correct answer. We find that: 1) the performance gains increase then decrease by rising the inherent difficulty; 2) performance gains increase with the number of steps; and 3) performance increases with the prior probability. Based on these properties, we also develop ways to boost the effectiveness of "More Agents".

Considering that each input remains the same when we increase the number of agents, the sampling phase can be optimized to reduce the cost. Nonetheless, such a challenge of escalating costs commonly exists in works requiring multiple LLM calls Wang et al. (2023b); Du et al. (2023). This aligns with recent findings as discussed in Kapoor et al. (2024) that current benchmarks focus narrowly on accuracy, leading to costly agents. Additionally, the lack of adequate holdout sets and standardization in evaluation practices further complicates the development of cost-effective agents. Addressing these issues can enhance the practical efficiency of AI agents in real-world scenarios. We leave it as a future work to optimize.

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

## A  Detailed Experiment Settings

### A.1  Common Settings

In all experiments involving GPT-3.5-Turbo presented in Section 4, we utilize the model version `gpt-3.5-turbo-0613`. In Table 2, the notation GPT-4 corresponds to the model version `gpt-4-0613`. For the experiments conducted with GPT-3.5-Turbo in Section 6, we employ the model version `gpt-3.5-turbo-1106` with the JSON mode enabled. Similarly, GPT-4 in this context refers to `gpt4-1106-Preview` operating in JSON mode.

### A.2  Experiments on Arithmetic Reasoning Tasks

For the implementation of Agent Forest to arithmetic reasoning tasks within the GSM8K and MATH datasets, we execute the initial sampling phase by Algorithm 1. Samples are extracted from the responses by matching "boxed{{X}}", where "X" denotes a numerical value or a mathematical expression.

In the subsequent voting phase, to evaluate the similarity between samples, as outlined in Algorithm 1, we employ mathematical equivalence comparisons for each sample. The most probable sample is chosen as the final answer. This answer is subjected to a comparison of mathematical equivalence with the ground truth to ascertain the correctness of the result.

### A.3  Experiments on General Reasoning Tasks

For general reasoning tasks, as encountered in the MMLU and Chess datasets, Agent Forest is applied following Algorithm 1 during the sampling phase. Samples are extracted by matching the pattern "(X" or "(X)", where "X" corresponds to the options A, B, C, or D in MMLU task and the chessboard position in Chess task.

During the voting phase, we calculate similarity by counting the frequency of each option's occurrence within the samples. The most frequently occurring option is then chosen as the final answer. This selected answer is compared with the ground truth to determine the accuracy of the result.

### A.4  Experiments on Code Generation Task

In the code generation task, we apply the Agent Forest method to produce Python code using the HumanEval dataset. During the sampling phase, we extract Python code snippets from the model's responses.

In the voting phase, we compute the `BLEU` score using *sacreBLEU* Post (2018) to evaluate the similarity between each of the generated samples. The sample with the highest cumulative `BLEU` score is selected as the final answer. This method ensures that the final output is the most representative and accurate piece of code as determined by consensus through similarity scoring among the samples.

## B  Detailed Experiment Results

### B.1  Additional Accuracy Curves

In this section, we provide the accuracy curves of our experiments across various datasets when utilizing different LLMs. From these curves, we demonstrate that our method has the following properties:

- **Generalizability.** By using our method standalone, the performance can be generally enhanced by increasing the ensemble size.

- **Compatibility.** Our method can generally enhance other methods by increasing the ensemble size.

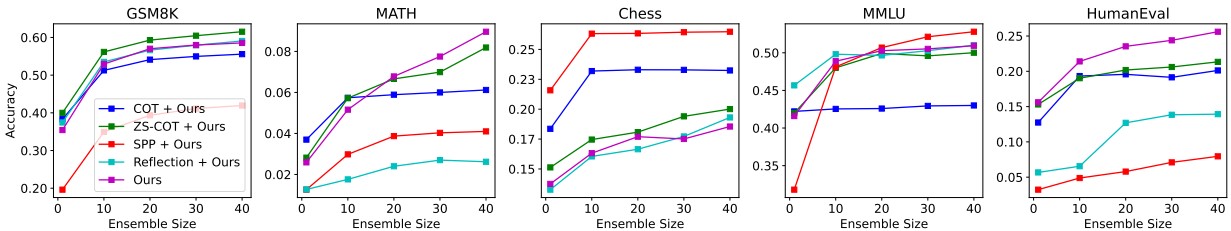

Figure 8: Accuracy curves across various datasets using the Llama2-13B model.

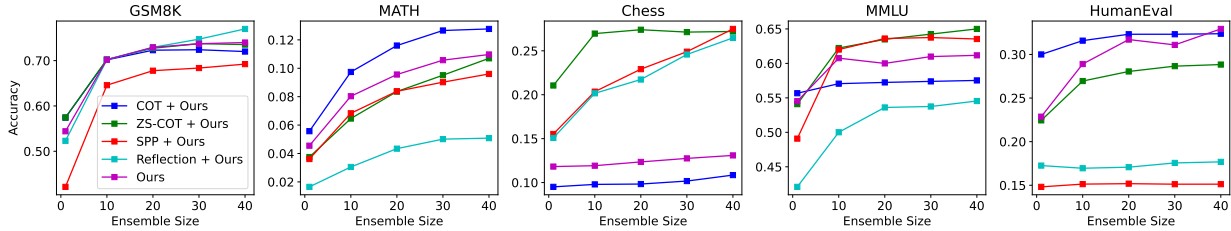

Figure 9: Accuracy curves across various datasets using the Llama2-70B model.

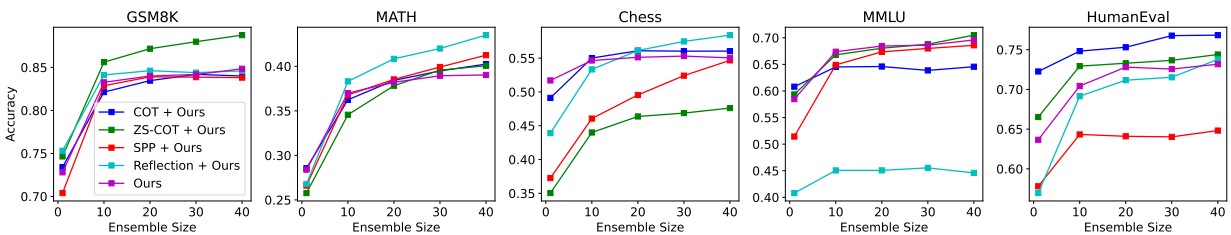

Figure 10: Accuracy curves across various datasets using the GPT-3.5-Turbo model.

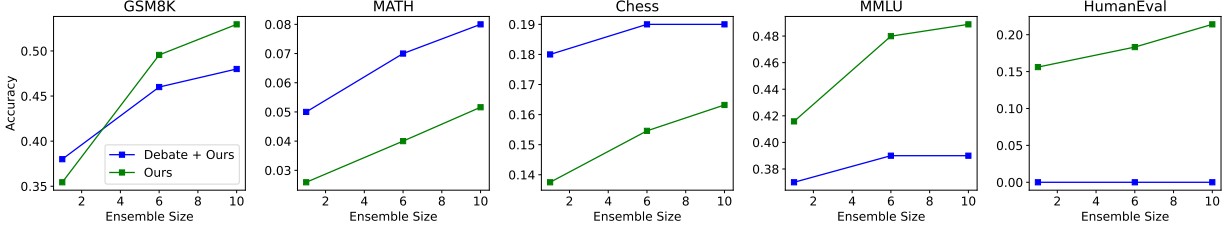

Figure 11: Debate accuracy curves across various datasets using the Llama2-13B model.

## B.2 Statistical Test Results

In this section, we performed a one-way ANOVA to determine if there are significant differences in accuracy across different ensemble sizes (1, 10, 20, 30, 40), across 10 independent runs. The detailed p-values are provided in Table 7, where all p-values are less than 0.05, demonstrating that the differences in accuracy between ensemble sizes are significant.

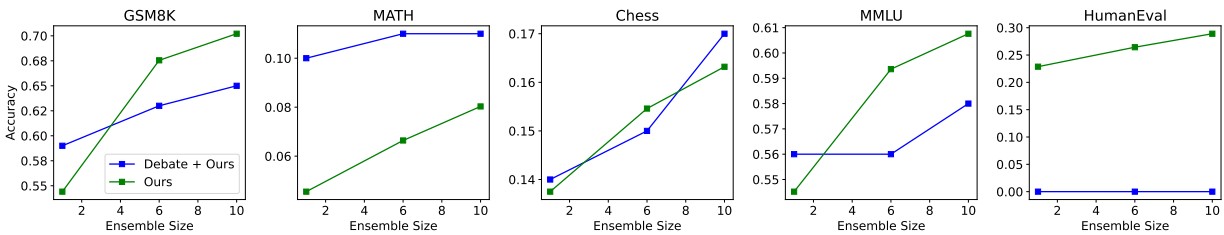

Figure 12: Debate accuracy curves across various datasets using the Llama2-70B model.

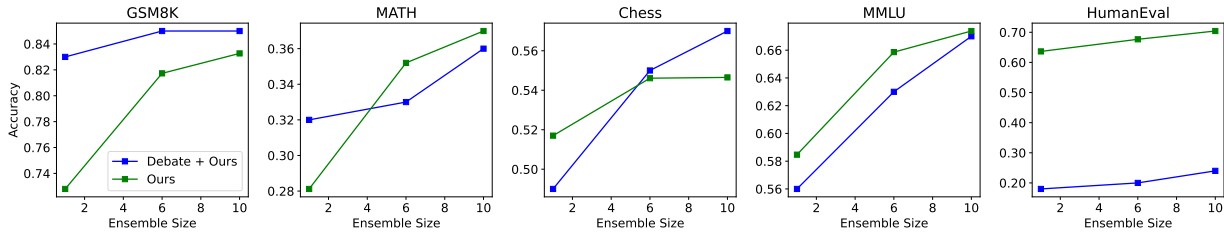

Figure 13: Debate accuracy curves across various datasets using the GPT-3.5-Turbo model.

Table 7: Statistical test results (p-values) of different LLMs on different datasets. One-way ANOVA is performed where ensemble sizes are (1, 10, 20, 30, 40).

| LLMs | GSM8K | MATH | Chess | MMLU | HumanEval |
|------|-------|------|-------|------|-----------|
| Llama2-13B | 1.56e-44 | 9.11e-32 | 1.97e-18 | 7.86e-38 | 5.46e-8 |
| Llama2-70B | 1.93e-43 | 2.16e-29 | 1.09e-9 | 1.06e-26 | 5.24e-7 |
| GPT-3.5-Turbo | 6.03e-30 | 9.66e-39 | 4.62e-24 | 2.46e-39 | 7.47e-13 |

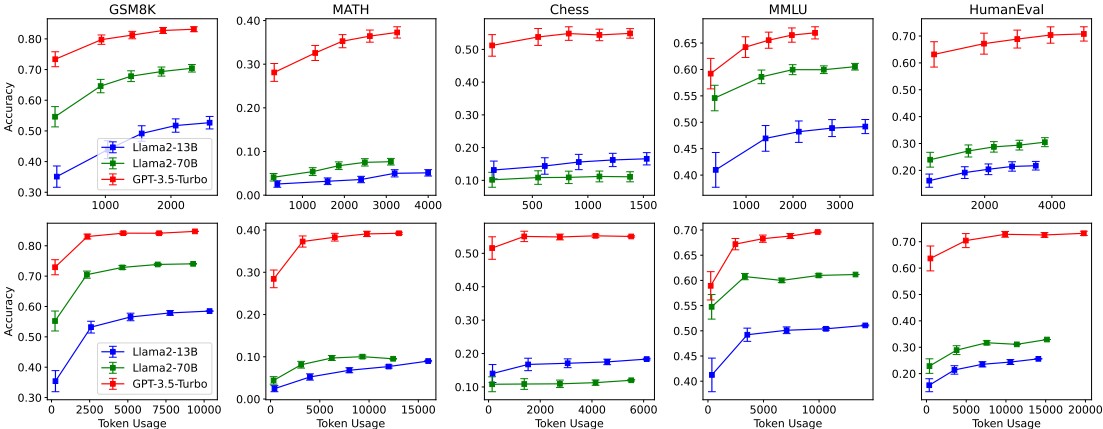

Figure 14: Token usage vs. accuracy.

## B.3 Token Usage vs. Accuracy

Figure 14 shows the tradeoff between token budget and accuracy, where the x-axis represents the token budget and the y-axis represents the accuracy.

