# OpenReview forum: "More Agents Is All You Need"
_TMLR — Accepted by TMLR_

### Review · Reviewer_EgWB · 2024-06-23

**Summary Of Contributions:**

# contributions
- A systematic exploration of how ensembling LLMs (using parallel answering and majority vote selection) can improve LLM performance across a variety of task domains including multiple choice question answering and code generation.
	- Includes using BLEU scores to implement majority voting for open-ended text-based outputs (specifically code).
- An analysis of task difficulty components and how they influence ensemble scaling in a toy case study.

# new knowledge
- Simply using multiple agents together with majority voting can substantially improve LLM performance on multiple choice reasoning and code generation tasks. LLM performance scales in (simple) ensemble size.
- Ensembling benefits (mostly) complement reasoning methods that augment base LLM predictions (like Chain-of-Thought prompting), including multi-agent approaches (like Debate).
- Weaker LLMs gain more from this ensembling than stronger LLMs.
- Harder tasks benefit more than easier tasks.

**Audience:**

Yes

**Broader Impact Concerns:**

The broader impact statement is generically about LLMs, but this is because the work is a very general technique.

**Claims And Evidence:**

No

**Requested Changes:**

# critical
- Add error bars to all plots and tables, along with statistical tests of differences among methods (where multiple runs are available).
	- The current evidence is favorable, but without variance measures (LLMs are often high variance) it is hard to know how strong the results are.
- Include reporting on token costs vs performance for the experiments.
	- This would directly address the question of how "smart" ensembling is: does it get more or less improvement in performance when consuming the same fixed inference (token) budget?
	- This will help characterize the scaling trade-offs being made with ensembles. Even if ensembling is more expensive for a given performance level it is still valuable knowledge to contribute and offers a meaningful baseline that other methods will need to show they outperform (in the future).
	- For example: a scatter plot showing the token usage vs performance for the different ensemble sizes and combined techniques.
		- This would be a plot using the data from something like Table 2 or Figure 3.
- Apply the stepwise and hierarchical methods to one or more of the main domains in the text (not just the toy domain)
	- This would make the paper an obviously very strong result and reinforce the value of the difficulty facet analysis.


# strengthen
- "Additionally, the experimental results indicate a correlation between the efficacy of the performance improvements and the difficulty of the problems addressed."
	- Be explicit: there are greater performance improvements on more difficult tasks and for weaker baseline models.
	- No reason to be vague with terms like "correlation" (positive or negative?) or to leave readers in suspense.
- Figure 1
	- How do the larger ensembles differ to the baseline in token costs?
	- This additional information would help when considering what a "fair" comparison might be for the amount of inference used by different methods.
- Table 2
	- How many agents are in the "Ours" ensemble? 40? This was suggested but not explicitly stated in the text (since it mentions scaling up the ensemble size while this is a single number)
- Figure 3
	- This would be a stronger figure if it included all the ensembles of size 1 to 10. That is the point of steepest increase of the curve, so the extra detail would help understand how rapidly the benefits of ensembling diminish. This is clearly sharp in GSM8K and MMLU, but not obviously so in MATH, Chess, or HumanEval.
	- I ask as this would better characterize how ensembling helps on different tasks, informing others who may want to repeat these experiments with smaller scale on newer domains.
- Section 5.1
	- Any idea why Chess shows such small gains (1% to 4%) compared to other domains?
		- Can this be related to the difficulty facets of steps or prior probability of success (or inherent difficulty, though I'm not sure how to measure that)?
	- Table 3 shows this domain to be challenging for "Ours" across base models.
- Figure 4
	- Please include the default T and p values used in the caption for easy reference.
- Section 6
	- It may be worth explicitly mentioning in the text that the weaker base models have greater performance gains. This is a valuable result.

**Strengths And Weaknesses:**

# strengths
- Presents a simple method with strong results across many domains.
- Analyzes the relative benefits of ensembling by task difficulty and base model scale / capability, providing new knowledge about how LLM inference structures scale relative to tasks.
- Provides clear evidence of the value of majority voting ensembles to complement other methods, showing useful generality. General, simple methods are welcome additions to the LLM design toolkit.

# weaknesses
- Statistical rigor: there are no error bars in plots/tables or statistical tests of differences.
	- Reporting statistical variability is important to quantify the benefits of the method. My concern is not that the magnitude be large, but that the tests are missing altogether.
- Lacks a cost-matched comparison of methods.
	- Ensembling incurs greater inference costs (measurable as number of tokens inferred) from a base LLM. This means adding more agents to an ensemble is an "unfair" comparison to models that use a fixed, smaller budget (like bigger LLMs).
	- Ideally the analysis would include reporting on model performance vs tokens produced. How do ensembling performance gains scale relative to token consumption? Where do the other methods compared fall along those same dimensions?
	- This will provide a more "fair" comparison and better clarify what the cost-benefit trade-offs are to ensembles. They clearly are beneficial in inference _time_ requirements: majority voting can be fully parallelized, unlike techniques like chain-of-thought or debate. This may also be useful as a comparison, though of secondary concern compared to the token consumption requirements.
- The analysis of difficulty facets feels unrelated to the main paper.
	- It would help to show the step-wise and hierarchical methods can apply to at least one of the base problem domains (that is not the toy example).

---

> ### Author Response · Authors · 2024-07-19
>
> Thank you for your thoughtful feedback. We addressed your concerns point by point.
>
> > **W1: Statistical rigor: there are no error bars in plots/tables or statistical tests of differences.**
>
> In the revised version, we have added error bars to the results (Figure 1, 3, 6 and Table 2). We observe that the errors of our method decrease as the ensemble size increases. This is due to the majority-voting mechanism, which ensures that the final result has the highest probability and reduces the errors.
>
> > **W2: Include reporting on token costs vs performance for the experiments.**
>
> We record the token usage for different methods, with this table presenting the token usage for a single agent (GPT-3.5). Given that our method scales up by increasing the ensemble size, the token usage increases proportionally with the number of agents or when combined with other methods. When addressing specific tasks, one can trade a higher token budget for improved performance.
>
> | Method                  | GSM8K       | MATH | Chess | MMLU | HumanEval |
> |-----------|--------------------------|---------|--------|---------|-----------|
> | Ours (single agent) | 235 ± 54 | 326 ± 131 | 138 ± 14 | 247 ± 91 | 495 ± 120 |
> | COT  | 261 ± 63 | 360 ± 134 | 284 ± 76 | 330 ± 110 | 330 ± 116 |
> | ZS-COT  | 228 ± 56 | 305 ± 122 | 132 ± 12 | 230 ± 92 | 305 ± 132 |
> | SPP | 341 ± 85 | 471 ± 161 | 363 ± 35 | 428 ± 123 | 501 ± 125 |
> | Reflection | 214 ± 45 | 281 ± 96 | 146 ± 13 | 218 ± 80 | 338 ± 110 |
>
> In the revised version of the paper, we have added a new section to discussing token costs.
>
> > **W3: Apply the stepwise and hierarchical methods to one or more of the main domains in the text.**
>
> Thank you for your suggestion. The purpose of this experiment is to verify whether the step-by-step method can achieve further improvements and to understand the patterns of such improvements. Therefore, we have designed a simple arithmetic task with clear structure and steps to facilitate analysis. For more complex tasks, related work has conducted similar attempts in other scenarios, such as [1, 2], where complex problems are broken down into multiple steps, theoretically leading to improvements. We hope to explore this in our future work.
>
> > **Strengths**
>
> We have revised the original manuscript to address the strengths highlighted in the  feedback. To facilitate your review, all new content has been highlighted in **red**, and any deleted content has been struck through in **blue**.
>
> References:
>
> [1] Deliberate Problem Solving with Large Language Models
>
> [2] Graph of Thoughts: Solving Elaborate Problems with Large Language Models

---

> > ### Comment · Reviewer_EgWB · 2024-07-19
> >
> > > In the revised version, we have added error bars to the results (Figure 1, 3, 6 and Table 2)
> >
> > Thank you! The differences are clearly large. If possible it would help to also include statistical tests showing these differences are significant. Also, please indicate what is shown with the error bars (perhaps also in the captions), I'm assuming it is reporting standard errors, but that was not made clear.
> >
> > > Given that our method scales up by increasing the ensemble size, the token usage increases proportionally with the number of agents or when combined with other methods.
> >
> > Thank you for providing this. My question was about token usage vs accuracy. Can you please show that data as well (even if only in the appendix)?
> >
> > My interest was understanding the Pareto front of token budget vs accuracy tradeoffs for these methods. Right now a reader has to compile that information and do the analysis manually. Providing that information would facilitate: "trade a higher token budget for improved performance."

---

> ### Author Response · Authors · 2024-07-22
>
> Thank you for your thoughtful feedback. We addressed your concerns point by point.
>
> > **Q1: If possible it would help to also include statistical tests showing these differences are significant.**
>
> Thank you for your suggestion. In Figures 1, 3, 6, and Table 2, the error bars represent standard errors. We have indicated this in the revised version.
>
> We also performed a one-way ANOVA to determine if there are significant differences in accuracy across different ensemble sizes (1, 10, 20, 30, 40), across 10 independent runs. The detailed p-values are provided below, where all p-values are less than 0.05, demonstrating that the differences in accuracy between ensemble sizes are significant. We have included this part in the revised version in Appendix B.2.
>
> | LLMs                  | GSM8K       | MATH | Chess | MMLU | HumanEval |
> |-----------|--------------------------|---------|--------|---------|-----------|
> | Llama2-13B | 1.56e-44 | 9.11e-32 | 1.97e-18 | 7.86e-38 | 5.46e-8 |
> |  Llama2-70B  | 1.93e-43 | 2.16e-29 | 1.09e-9 | 1.06e-26 | 5.24e-7 |
> | GPT-3.5-Turbo  | 6.03e-30 | 9.66e-39 | 4.62e-24 | 2.46e-39 | 7.47e-13 |
>
> > **Q2: My question was about token usage vs accuracy. Can you please show that data as well (even if only in the appendix)?**
>
> A2: Thank you for your suggestion. To visualize the tradeoff between token budget and accuracy, we have plotted a scatter plot where the x-axis represents the token budget and the y-axis represents the accuracy in the revised version in Appendix B.3.

---

### Review · Reviewer_EUQF · 2024-07-05

**Summary Of Contributions:**

This paper mainly studied how to use LLMs to generate multiple results and then ensemble it to get better performance. Experimental results demonstrate the effectiveness of the proposed method.

**Audience:**

Yes

**Broader Impact Concerns:**

This paper does not include broader impact concerns.

**Claims And Evidence:**

Yes

**Requested Changes:**

Please see my above comments.

**Strengths And Weaknesses:**

**Strengths**:

This paper studied the capability of integrating multiple results from LLMs.

**Weaknesses**:

I think this paper just simply generate multiple outputs by using LLMs and then ensemble results. The core idea of this paper is still the thinking of ensemble learning, and I don't see any other contributions in term of methods. Get multiple results and then using voting strategy to improve performance is not a novel idea. And this paper just uses LLMs to get results. Moreover, I think this paper just applied prompting to LLMs but claim it as LLM-based agents. I think this is an over-claim. Actually, this paper just use LLMs or LLMs plus different prompting strategies. I didn't see any unique insights in agents.

---

> ### Author Response · Authors · 2024-07-19
>
> Thank you for your thoughtful feedback. We addressed your concerns point by point.
>
> > **W1: Lack of Novelty in Method and  Over-claim of LLM-based Agents.**
>
> Thank you for your insightful feedback on our paper.  We respectfully disagree with your assessment and would like to provide further clarification on our contributions.
>
> Firstly, the methodology presented in our paper is both simple and effective, and it is the first to apply the ensemble learning concept to LLM-based agents.  As demonstrated in Table 4, our approach outperforms other so-called novel methods across all five tasks and three LLMs , showcasing its effectiveness.  This simplicity and efficiency make our method more user-friendly and convenient, eliminating the need for complex interaction structures or additional prompts.
>
> Secondly, in our revised version, we have included error bars to better reflect the robustness of our method's performance.  Additionally, we have incorporated a cost analysis, allowing users to choose the appropriate range based on their specific needs.
>
> Regarding the reviewer's comment on our use of LLMs and the claim of LLM-based agents, we would like to emphasize that our approach indeed embodies a broader definition of an agent.  Our methodology not only integrates LLMs but also can be combined with other LLM-based agents.  The incorporation of LLM-based agents within our framework clearly qualifies it as an LLM-based agent approach.  By framing LLMs as agents, we provide a structured approach that harnesses their potential in a coordinated and effective manner, extending beyond traditional prompting methods.
>
> We hope this clarification addresses your concerns and highlights the significant contributions of our work in advancing the state-of-the-art in LLM applications.

---

### Review · Reviewer_gVUG · 2024-07-07

**Summary Of Contributions:**

This paper studies the effect of increasing the ensemble size of a variety of LLM sampling methods on performance. It proposes a simple majority vote strategy which leverages the similarity of generations to aggregate proposed outputs in natural language tasks with combinatorially large output spaces.

**Audience:**

Yes

**Claims And Evidence:**

Yes

**Requested Changes:**

See weaknesses.

**Strengths And Weaknesses:**

### Strengths

- The proposed method is simple and clearly explained.

- The evaluations seem reasonably comprehensive, covering three categories of reasoning tasks and considering both Llama and GPT models of varying sizes.

- The majority-vote trick is complementary and easy to incorporate into a variety of existing methods

- Even the simplest variant of ensembling to which the method is applied, self-ensembling, produces notable improvements via the majority-vote strategy.

- The sweep over ensemble sizes in e.g. Figure 4 is sufficient to conclusively show benefits to scaling the number of ensemble members as well as limitations where the performance gains saturate.

### Weaknesses

- The percentage improvements in Figure 6 are quite small and there are no error bars to indicate whether the trends observed are real or due to random noise. I would like to see a significance test done on the claim that the improvement at I=400 is significantly smaller than I=100 or 200.

- More generally, it would be helpful to see error/standard deviation bars just to get a sense of the significance of the results.

- While the additional investigations into the effect of different aspects of problem difficulty on the performance gains from the method were helpful, I would have liked to see the qualitative investigations focus more on explaining why there are such disparate improvements across the different domains. For example, the chess dataset sees relatively minimal gains, especially on Llama models — is this because the dataset is too hard? Or is it because of the output and reasoning structure that makes it a bad fit for the majority-vote similarity measure?

- In section 6, it is argued that harder tasks see greater benefit from the method, but it isn’t clear to me whether this statement is a generic truth or in part an artifact of how relative performance gain is computed (for example, going from 25 to 50% accuracy is a greater improvement under this metric than improving from 50 to 99%). If instead of measuring gain as relative increase in accuracy, we instead modeled it as relative decrease in error, would the harder tasks still demonstrate greater improvement?

- The term ‘prior probability’ is a bit confusing — does this refer to a uniform uninformed Bayesian prior over feasible answers to a subproblem, or does it implicitly refer to the LLM’s probability of generating the correct answer without following chain of thought reasoning first?

### Questions

- Did you consider alternative choices for the majority-vote clustering metric? Do different choices give different performances?

- The main difference between this approach and that of Wei et al. is its broader application of majority-vote beyond CoT reasoning tasks to code generation and to intermediate outputs at CoT steps. While the latter does help reduce the risk of compounding errors in long-range reasoning tasks, it also seems like it could run the risk of prematurely committing to an incorrect reasoning step if one type of reasoning error produces outputs that are all extremely similar based on their BLEU scores. Could the authors comment on when / if ever such a situation might occur?

---

> ### Author Response · Authors · 2024-07-19
>
> Thank you for your thoughtful feedback. We addressed your concerns point by point.
>
> > **W1:  Small percentage improvements and lack of error bars**
>
> In the revised version, we have added error bars to Figure 1, 3, 6 and Table 2. It can be observed that the improvement at I=400 is less significant compared to I=100 or I=200.
>
> > **W2: Explanation for disparate improvements across domains. For example, the chess dataset sees relatively minimal gains, especially on Llama models...**
>
> The slower performance improvement is due to the increased difficulty of the Chess dataset. Both the Chess and MMLU datasets are reasoning tasks involving multiple-choice questions. For MMLU, it is a single-choice task (1 out of 4), whereas Chess involves multiple choices (selecting feasible positions from the entire chessboard). The prior probability of generating the correct answer in Chess is significantly lower than in MMLU. Consequently, more results are needed during majority voting to highlight the correct answer, leading to slower performance improvement.
>
> > **W3: Clarification on benefit of method for harder tasks.  Whether the benefit is a generic truth or in part an artifact of how relative performance gain is computed.**
>
> The benefits are both a general fact and a result of calculating relative performance gains. This is because calculating relative performance gains can reflect the benefits. For instance, consider a 30% accuracy improvement in two different scenarios: from 10% to 40% and from 50% to 80%. In the first scenario, an accuracy of 10% means that, on average, 10 attempts are needed to get the correct answer. Improving this to 40% reduces the number of attempts to 2.5, effectively decreasing the attempts by 7.5. In contrast, for the second scenario, the reduction is only 2 - 1.25 = 0.75 attempts. This illustrates that the greater the relative performance gain, the fewer attempts are needed to obtain the correct answer, which is beneficial for users. Similarly, the relative reduction in calculation errors has corresponding benefits and observable results. This does not affect the generality of the benefits; it only concerns what the specific benefits are.
>
> > **W4: Clarification on 'prior probability' term.**
>
> Here we aim to quantify the difficulty of the task. The term "prior probability" refers to a uniform, uninformed Bayesian prior over the feasible answers to a subproblem, which can be determined by examining the solution space. As the solution space expands, the task inherently becomes more challenging.
>
> > **Q1: Consideration of alternative majority-vote clustering metrics.**
>
> We aim to integrate multiple agents using the simplest possible method, which facilitates implementation and usage while achieving satisfactory results. Other methods, such as those described in [1], involve further filtering of multiple agent results using LLM, leading to additional improvements. However, this is not the focus of our work. Our primary focus is to determine whether such a simple method can achieve improvements across a wide range of domains, quantify the extent of these improvements, and understand the underlying patterns.
>
> > **Q2:  Risk of Premature Commitment to Incorrect Reasoning. Could the authors comment on when / if ever such a situation might occur?**
>
> This may occur when the task is particularly difficult. However, the intermediate steps required to solve a problem are theoretically simpler than solving the problem directly. If errors occur in the intermediate steps, then the probability of errors in directly solving the problem is even higher. Therefore, theoretically, using majority voting on the intermediate steps is more effective than applying majority voting directly on the CoT (Chain of Thought) results.
>
> References:
>
> [1] Ask One More Time: Self-Agreement Improves Reasoning of Language Models in (Almost) All Scenarios

---

### Decision · Action_Editor_tALM · 2024-09-28

**Recommendation:** Accept as is

**Comment:**

While the methods proposed in this paper are not particularly novel themselves (ensembling being a well-studied method in ML for decades), their application to agents and demonstration of effectiveness make this paper interesting and useful to the TMLR audience and will inform future work in this space. The authors clarified questions and improved the paper based on reviewer suggestions.

The one recommendation I have for the authors is to include some concurrent work along similar lines (like https://arxiv.org/abs/2407.01502) in the discussion so that the readers may gain a more comprehensive understanding of the core ideas in the paper.

**Audience:**

Given the rising interest in building autonomous agents on top of LLMs and other generative models, this paper will be interesting to a significant proportion of TMLR audience.

**Claims And Evidence:**

This paper empirically investigates the effect of simple ensembling methods for agents, such as sampling and majority voting. The experiments provide comprehensive evidence that such techniques work well and can be complementary to other methods of enhancing LLMs and decision making for agents.